# Comparison of Clinical Features, Complete Blood Count Parameters, and Outcomes between Two Distinct Waves of COVID-19: A Monocentric Report from Italy

**DOI:** 10.3390/healthcare10122427

**Published:** 2022-11-30

**Authors:** Sara Solveig Fois, Elisabetta Zinellu, Angelo Zinellu, Michela Merella, Maria Carmina Pau, Ciriaco Carru, Alessandro Giuseppe Fois, Pietro Pirina

**Affiliations:** 1Department of Medicine, Surgery and Pharmacy, University of Sassari, 07100 Sassari, Italy; 2Clinical and Interventional Pulmonology, University Hospital of Sassari (AOU), 07100 Sassari, Italy; 3Department of Biomedical Sciences, University of Sassari, 07100 Sassari, Italy

**Keywords:** COVID-19, pandemic waves, CBC, inflammation, biomarkers

## Abstract

Background: Since the beginning of the SARS-CoV-2 pandemic, the ability to predict the trajectory of the disease has represented a major challenge for clinicians. There is recent evidence that complete blood cell count (CBC)-derived inflammation indexes have predictive value in COVID-19. We aimed to describe any changes in the clinical features, CBC-derived ratios, and outcomes of patients admitted to our hospital across two temporally distinct waves. Methods: We retrospectively assessed and compared the clinical characteristics and blood cell count values of patients hospitalized during the second and fourth waves of COVID-19, and explored any outcome differences in terms of the level of respiratory support required and transfer to intensive care. Results: We observed that fourth-wave patients were older, less male-predominant, and carried more comorbidities compared to the second-wave patients but, nevertheless, experienced more favorable outcomes. A strong internal correlation was documented for both waves between outcomes and CBC-derived ratios, with the fourth-wave cases displaying lower admission values of the neutrophil-to-lymphocyte ratio (NLR), derived NLR (dNLR), platelet-to-lymphocyte ratio (PLR), and systemic inflammation index (SII). No significant differences were found for lymphocyte-to-monocyte ratio (LMR), systemic inflammation response index (SIRI), and aggregate index of systemic inflammation (AISI). Conclusions: We observed that both admission values of CBC-derived indexes and adverse respiratory outcomes decreased from the second to the fourth wave of COVID-19. These data represent a contribution to the existing knowledge on the role of CBC-derived indexes as a potential tool to help clinicians to quickly differentiate in-hospital patients at increased risk of serious illness and death.

## 1. Introduction

Over two and a half years have passed since the World Health Organization declared the Coronavirus Disease 2019 (COVID-19) a global pandemic, the spread of which has impacted all aspects of life and has negatively affected many areas of healthcare, medical activity, and research worldwide [1,2,3,4,5,6,7,8].

The dynamics of COVID-19 have been heterogeneous across countries, with several differences in incidence, infection, and mortality rates both spatially and temporally between distinct epidemic waves [9,10,11]. Much of this variation has been correlated with containment measures, socioeconomic status, population structure and density, healthcare system responses, and most recently, vaccine acceptance and efficiency [12,13,14,15,16,17,18,19,20]. In Italy, different lethality rates have been observed across different areas of the country from the early days of the first outbreak. There is evidence that the lack of shared health management policies and clinical care pathways between regional hospital networks has played a crucial role in the diversity of pandemic outcomes [21]. The problem has been further complicated by the emergence and spread of new variant strains of the virus, which have added more complexity to a disease for which the clinical course is difficult to describe and predict. In fact, while most patients experience mild respiratory symptoms and recover without any special treatment, some can become critically ill and may not survive even with intensive therapy [22]. 

These premises highlight the need to expand the prognostic factor landscape of COVID-19, with the goal of improving risk stratification at the early stages of the illness by identifying risk factors for progression to severe disease [23]. In this regard, certain hematological parameters and their derived ratios have been associated with the proinflammatory response typical of COVID-19-related organ failure and mortality. This finding has generated great interest in employing these indexes as biomarkers to establish prognosis and appropriate level of care, particularly because they are cheap and easily evaluated through routine blood tests [24,25]. In particular, the neutrophil-to-lymphocyte ratio (NLR), derived NLR (dNLR), systemic inflammation index (SII), and aggregate index of systemic inflammation (AISI) have so far produced the most promising evidence in terms of their potential use for early risk stratification of COVID-19 patients [26,27,28,29,30,31,32,33,34,35,36,37,38,39,40].

In our previous study, we showed that SII predicted in-hospital mortality in patients admitted during the first wave of COVID-19 [41,42]. In the present study, we sought to report on the clinical and hematological characteristics of patients admitted during the second and fourth waves, and to investigate any differences between these waves in terms of the level of respiratory support required and transfer to intensive care. We aimed to describe any significant changes in the admission levels of hematological ratios, and to discuss the potential practical applications of these indicators in the context of the need for improvement of present and future pandemic management. 

## 2. Materials and Methods

### 2.1. Study Design

We conducted a retrospective study of medical records from a population of 342 consecutive hospitalized cases of COVID-19, including 182 patients from the second wave and 160 patients from the fourth wave. All patients were admitted to the respiratory disease unit of the University Hospital of Sassari, Italy, a district hospital serving a geographical area with a population of about 200,000. Data are presented as a comparison between the second and fourth waves in terms of clinical characteristics, hematological data, and clinical outcomes, defined as the level of respiratory support required and the rate of transfer to intensive care. We did not extend our study to the third pandemic wave because it produced a negligible number of hospitalizations in our target area.

### 2.2. Inclusion Criteria 

To be included in the study, patients had to return a positive SARS-CoV-2 polymerase chain reaction (PCR) test result and have had a complete blood cell count on the first day of admission. Criteria for admission to the ward included hypoxia or worsening oxygen requirement, as well as patients at high risk for respiratory complications due to advanced age or underlying comorbidities. 

### 2.3. Data Collection

All demographic, clinical, and laboratory information was retrospectively collected from the electronic medical records. The Charlson Comorbidity Index was used as a weighted score of the patients’ comorbidities [43]. The following hematological parameters were evaluated: hemoglobin (HGB), red blood cell count (RBC), red blood cell distribution width (RDW), white blood cell count (WBC), neutrophils, lymphocytes, monocytes, platelet count (PLT), and mean platelet volume (MPV). We then calculated the following CBC-derived indexes of systemic inflammation: NLR (neutrophil-to-lymphocyte ratio), derived NLR (neutrophils/(white blood cells − neutrophils)), LMR (lymphocyte-to-monocyte ratio), PLR (platelet-to-lymphocyte ratio), SII ((neutrophils × platelets)/lymphocytes), SIRI ((neutrophils × monocytes)/lymphocytes), and AISI ((neutrophils × monocytes × platelets)/lymphocytes). We included information regarding the level of respiratory support received during hospitalization, and distinguished patients who received oxygen supplementation only from patients who required respiratory support either with continuous positive airway pressure (CPAP) or with noninvasive mechanical ventilation (NIMV). Surveillance for each patient continued throughout the duration of their stay on the respiratory unit until either home discharge, transfer to another ward, transfer to intensive care, or death on the ward. 

The study was conducted in accordance with the Declaration of Helsinki and was approved by the ethics committee of the University Hospital (AOU) of Cagliari (PG/2020/10915).

### 2.4. Statistics

For variables with continuous distributions, the Kolmogorov–Smirnov test was applied. Because none of the data were normally distributed, the results were expressed as median values (median and interquartile range (IQR)). 

Between-group differences in demographic, clinical, and laboratory data were compared using the Mann–Whitney test. Differences between categorical variables were evaluated via chi-squared test. Correlations between variables were estimated using Spearman’s rank correlation. Statistical analyses were performed using MedCalc for Windows, version 20.109 bit (MedCalc Software, Ostend, Belgium).

## 3. Results

A total of 342 patients were involved in the study. From 12 October 2020 to 26 January 2021, a period corresponding to the second wave of the COVID-19 pandemic in Italy, 182 confirmed cases were admitted to our respiratory unit ward. The second group of 160 patients was admitted during the fourth wave, starting from 20 December 2021 and continuing until 22 April 2022. 

### 3.1. Demographic and Clinical Characteristics

The demographic and clinical characteristics of both groups are shown in Table 1.

The second-wave cases consisted of 123 males and 59 females. The median age was 72 (IQR: 62–83) years, and the median body mass index (BMI) was 27.2 (IQR 25–29.2). The weight of comorbidities resulted in a median Charlson Comorbidity Index score of 1 (IQR: 0.0–2.0). PaO_2_/FiO_2_ (P/F) ratio values on admission showed an IQR of 145–310, with a median value of 225. The fourth-wave cases included more women than were observed in the second wave (74 vs. 59) and 86 men, *p*-value = 0.009. The patients were also significantly older (78.5 years, IQR 67–84, *p*-value = 0.012) and had a lower BMI (median 25, IQR 22.5–29.4, *p*-value = 0.0497). The Charlson Comorbidity Index was twice as high compared to that observed in the second-wave group (median 2.0, IQR 0.5–3.0, *p*-value = 0.00002). There were no significant differences between the two groups in smoking status and P/F value on admission.

### 3.2. Hematological Characteristics

Hematological characteristics are shown in Table 2. 

A significant difference between the two groups was observed in the blood cell count values. The fourth-wave patients had significantly lower values of HGB (*p*-value < 0.0001) and RBC (*p*-value 0.05), while no significant difference was observed in RDW. The correlation between HGB and pathology (rho = −0.225, *p*-value < 0.0001) remained significant even after correcting for sex-related confounding factors (rho = −0.204, *p*-value = 0.0002). The fourth-wave patients also showed higher values of monocytes and lymphocytes compared to the second-wave patients (*p*-value < 0.05). There were no significant differences in WBC and neutrophils. Platelet levels were also similar, although in the fourth-wave group a higher MPV was observed (*p*-value < 0.0001). Second-wave patients showed higher values of NLR (median 8.50, IQR 4.15–14.94; vs. median 6.79, IQR 3.20–12.43, *p*-value = 0.029), dNLR (median 4.96, IQR 2.69–8.26, vs. median 3.81, IQR 1.91–6.58, *p*-value = 0.009), PLR (median 290, IQR 168–447 vs. median 236, IQR 117–377, *p*-value = 0.004), and SII (median 1899, IQR 778–3734, vs. median 1229, IQR 602–3096, *p*-value = 0.01). In contrast, there were no significant between-wave differences in LMR, SIRI, and AISI.

### 3.3. Clinical Endpoints

Clinical endpoints are depicted in Table 3. 

In the second wave, 42.5% (76/179) of patients required oxygen supplementation during their stay, a number that rose to 59.1%. (94/159) in the fourth wave, although the difference did not reach statistical significance (*p*-value = 0.08). In contrast, the proportion of patients requiring high-dependency-level care to manage respiratory failure—either with CPAP or NIMV—significantly decreased from 43.6% in the second wave to 23.9% in the fourth wave (*p*-value = 0.008). Transfer to intensive care also markedly dropped from the second (13.1%, 23/175) to the fourth wave (4.1%, 6/145), *p*-value = 0.01. Table 4 describes a significant internal correlation in both waves between the intensity of care required (measured as the number of patients treated with oxygen supplementation, CPAP/NIMV, or transferred to intensive care as described in Table 3) and all combined indexes, especially NLR, dNLR, and SII (*p*-value < 0.001).

## 4. Discussion

The present investigation compared clinical and hematological features of 342 patients admitted to a respiratory unit during the second and fourth waves of COVID-19, and investigated any correlation with outcomes in terms of the intensity of respiratory support required and the likelihood of transfer to an intensive care unit (ICU). 

Our study found that compared to the second wave, patients admitted to hospital during the fourth wave of COVID-19 were less male-predominant, notably older, and had a higher burden of underlying comorbidities. The same group experienced a significant drop in the need for respiratory support, as well as a marked reduction in ICU admissions. In retrospect, we observed a different pattern of laboratory characteristics, with the fourth-wave patients displaying lower values of NLR, dNLR, PLR, and SII on admission. 

On first impression, the shift towards hospitalization of patients that are older and more likely to suffer from comorbidities in the fourth wave may appear in contrast with the high levels of vaccination reached among the geriatric population in our country as of November 2021 [44]. However, that frail older adults are disproportionately affected by COVID-19 has been apparent since the very start of the pandemic [45]. Nonetheless, we observed that fourth-wave patients experienced a significant reduction in the need for either CPAP or NIMV, and were less likely to be transferred to intensive care. Apart from the obvious explanation that this group was protected by vaccination, there may be other elements contributing to this result, including virus-specific properties. Indeed, our first recruiting period appertained to patients most likely still infected with the ancestral strain variants EU1 and EU2, with the first variant of concern (VOC) Alpha (B.1.1.7) only appearing in Italy in February 2021 [46,47]. Similarly, we can assume that the vast majority of our fourth-wave patients were infected with the VOC Omicron (B.1.1.529), which was first detected in Italy in November 2021 and reached fixation within a few weeks [48,49]. Omicron presents a lower pathogenicity than prior SARS-CoV-2 variants [50], a circumstance that fits our study’s observation of a decreased risk of respiratory complications among fourth-wave patients. 

To the best of our knowledge, this is the first study to compare the levels of CBC-derived inflammation indexes of hospitalized patients from two different COVID-19 waves. In our previous investigation, we found increased values of CBC-derived ratios in severe COVID-19 patients of the first wave, as well as an independent association between SII and survival rate [42]. In the present study, we report an internal correlation between all the analyzed combined indexes (NLR, dNLR, LMR, PLR, SII, SIRI, and AISI) and the intensity of respiratory support required for both second- and fourth-wave patients, with NLR, dNLR, and SII reaching the most robust statistical significance (*p*-value < 0.001). 

CBC-derived indexes have gained increasing scientific interest over the last decade and are being explored as markers of inflammation in several other disorders, including pulmonary diseases such as chronic obstructive pulmonary disease, asthma, sleep apnea, and lung fibrosis [51,52,53,54]. Our findings provide a better comprehension of the pattern of these biomarkers in relation to the heterogeneous presentation of COVID-19 patients from a real-world experience of dissimilar pandemic waves. The implications of this work are informative in terms of the potential use of CBC-derived indexes as a low-cost tool for the physician for early detection and management of patients at high risk of developing respiratory complications and progression to severe disease. Indeed, our work describes the reduction of the admission values of NLR, dNLR, PLR, and SII from the second wave to the less severe, Omicron-driven fourth wave, an observation in accordance with the already well-established independent association between blood-cell-derived biomarkers and adverse outcomes in COVID-19 [26,27,28,29,30,31,32,33,34,35,36,37,38,39,40].

Our study has some limitations, the most obvious being that it was a retrospective study lacking external validity. Furthermore, as this was a single-center study based on the patients admitted to the respiratory disease unit of the University Hospital of Sassari, our findings may not be generalizable to other hospitals in our region. Finally, we could not retrieve any information on survival rates. On the other hand, we correlated our data with endpoints that have critical impact on the prognosis of COVID-19, such as the need for ventilatory support and transfer to ICU. This is also the first study to compare the prognostic role of CBC-derived indexes in two temporally distinct COVID-19 surges with different patterns of clinical and epidemiological characteristics.

## 5. Conclusions

We observed lower admission values of NLR, dNLR, PLR, and SII in COVID-19 patients admitted to our respiratory unit during the fourth wave of the pandemic, compared to the second wave. Overall, fourth-wave patients were older, less male-predominant, and had a higher comorbidity burden, but their risk of respiratory complications was decreased compared to second-wave patients. There was also a strong internal correlation between outcomes and CBC-derived ratios for both waves. These findings are in accordance with data from previous literature on the role of CBC-derived indexes for early risk stratification of COVID-19 patient. In the future, the development and validation of optimal cut-off scores for these biomarkers should be a focus of study. 

## Figures and Tables

**Table 1 healthcare-10-02427-t001:** Characteristics of the studied population.

	Wave 2 Median (IQR)/*n* (%)	Wave 4 Median (IQR)/*n* (%)	*p*-Value
Age (years)	72 (62–83) (*n* = 182)	78.5 (67–84) (*n* = 160)	0.012
Gender (male)	123 (67.6) (*n* = 182)	86 (53.7) (*n* = 160)	0.009
Body mass index (kg/m^2^)	27.2 (25–29.2) (*n* = 96)	25 (22.5–29.4 (*n* = 97)	0.0497
Smoking (yes)	48 (55.8) (*n* = 86)	31 (43.1) (*n* = 72)	0.08
Admission PaO_2_/FiO_2_ ratio	225 (145–310) (*n* = 180)	231 (181–303) (*n* = 139)	0.27
Cardiovascular disease (yes)	125 (68.7) (*n* = 182)	116 (72.5) (*n* = 160)	0.44
Respiratory disease (yes)	35 (19.2) (*n* = 182)	49 (30.6) (*n* = 160)	0.015
Kidney disease (yes)	15 (8.2) (*n* = 182)	23 (14.4) (*n* = 160)	0.07
Diabetes (yes)	45 (24.9) (*n* = 181)	38 (23.7) (*n* = 160)	0.81
Cancer (yes)	15 (8.2) (*n* = 182)	26 (16.3) (*n* = 159)	0.02
Autoimmunity (yes)	13 (7.2) (*n* = 181)	14 (8.7) (*n* = 160)	0.59
Charlson Comorbidity Index	1.0 (0.0–2.0) (*n* = 182)	2.0 (0.5–3.0) (*n* = 160)	0.00002

IQR: interquartile range; PaO_2_/FiO_2_ ratio: the arterial partial pressure of oxygen (PaO_2_) divided by the inspired oxygen concentration (FiO_2_).

**Table 2 healthcare-10-02427-t002:** Hematological characteristics of the studied waves.

	Wave 2 Median (IQR)	Wave 4 Median (IQR)	*p*-Value
HGB (g/dL)	13.5 (12.0–14.8) (*n* = 179)	12.1 (10.4–14.0) (*n* = 157)	0.000006
RBC (×10^12^ cells/L)	4.79 (4.25–5.22) (*n* = 179)	4.53 (3.87–5.01) (*n* = 157)	0.005
RDW (%)	14.9 (13.8–16.2) (*n* = 177)	14.8 (13.3–16.5) (*n* = 158)	0.33
WBC (×10^9^ cells/L)	8.30 (5.46–11.97) (*n* = 179)	8.05 (5.70–10.79) (*n* = 157)	0.94
Neutrophils (×10^9^ cells/L)	6.50 (4.20–10.42) (*n* = 179)	6.00 (4.28–9.20) (*n* = 157)	0.34
Lymphocytes (×10^9^ cells/L)	0.80 (0.50–1.20) (*n* = 179)	0.80 (0.60–1.50) (*n* = 157)	0.02
Monocytes (×10^9^ cells/L)	0.40 (0.21–0.50) (*n* = 179)	0.40 (0.30–0.60) (*n* = 157)	0.027
PLT (×10^9^ cells/L)	230 (169–292) (*n* = 178)	221 (159–293) (*n* = 157)	0.21
MPV (fL)	8.40 (7.90–9.20) (*n* = 178)	9.30 (8.20–10.30) (*n* = 158)	<0.000001
Combined indexes			
NLR	8.50 (4.15–14.94) (*n* = 179)	6.79 (3.20–12.43) (*n* = 157)	0.029
dNLR	4.96 (2.69–8.26) (*n* = 179)	3.81 (1.91–6.58) (*n* = 157)	0.009
LMR	2.23 (1.50–3.13) (*n* = 178)	2.31 (1.41–3.50) (*n* = 156)	0.66
PLR	290 (168–447) (*n* = 178)	236 (117–377) (*n* = 157)	0.004
SII	1899 (778–3734) (*n* = 178)	1229 (602–3096) (*n* = 157)	0.01
SIRI	2.80 (1.29–6.49) (*n* = 179)	2.60 (1.19–5.41) (*n* = 157)	0.46
AISI	632 (243–1615) (*n* = 178)	477 (251–1255) (*n* = 157)	0.22

IQR: interquartile range; HGB: hemoglobin; RBC: red blood cell count; RDW: red blood cell distribution width; WBC: white blood cell count; PLT: platelet count; MPV: mean platelet volume; NLR: neutrophil-to-lymphocyte ratio; dNLR: derived neutrophil-to-lymphocyte ratio; LMR: lymphocyte-to-monocyte ratio; PLR: platelet-to-lymphocyte ratio; SII: systemic inflammation index; SIRI: systemic inflammation response index; AISI: aggregate index of systemic inflammation.

**Table 3 healthcare-10-02427-t003:** Clinical endpoints of the studied waves.

	Wave 2 *n* (%)	Wave 4 *n* (%)	*p*-Value
Received oxygen therapy	76 (42.5) (*n* = 179)	94 (59.1) (*n* = 159)	0.08
Received CPAP/NIMV	78 (43.6) (*n* = 179)	38 (23.9) (*n* = 159)	0.008
Transfer to intensive care	23 (13.1) (*n* = 175)	6 (4.1) (*n* = 145)	0.01

CPAP: continuous positive airway pressure; NIMV: noninvasive mechanical ventilation.

**Table 4 healthcare-10-02427-t004:** Correlation between intensity of care and combined indexes of the studied populations.

	Wave 2 (*n* = 182)		Wave 4 (*n* = 160)	
	Rho	*p*-Value	Rho	*p*-Value
NLR	0.326	<0.001	0.368	<0.001
dNLR	0.315	<0.001	0.394	<0.001
LMR	−0.234	0.0017	−0.201	0.0123
PLR	0.232	0.0020	0.259	0.0011
SII	0.302	<0.001	0.323	<0.001
SIRI	0.263	0.0004	0.259	0.0011
AISI	0.236	0.0016	0.226	0.0046

NLR: neutrophil-to-lymphocyte ratio; dNLR: derived neutrophil-to-lymphocyte ratio; LMR: lymphocyte-to-monocyte ratio; PLR: platelet-to-lymphocyte ratio; SII: systemic inflammation index; SIRI: systemic inflammation response index; AISI: aggregate index of systemic inflammation.

## Data Availability

Not applicable.

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
