# Peer review of "Comparison of Clinical Features, Complete Blood Count Parameters, and Outcomes between Two Distinct Waves of COVID-19: A Monocentric Report from Italy"

_healthcare, 2022, doi:10.3390/healthcare10122427_

Round 1
Reviewer 1 Report
I would like to congratulate the authors on their work. This is a potentially significant research since it provides evidence for the pandemic's negative consequences on patient access to hospital treatment. However, there are a few aspects that must be addressed.
I recommend English to be slightly improved.
1. Abstract:
1.1 Please re-write the Abstract to conform to the requirements:” Background: Place the question addressed in a broad context and highlight the purpose of the study; Methods: Describe briefly the main methods or treatments applied. Include any relevant preregistration numbers, and species and strains of any animals used. Results: Summarize the article's main findings; Conclusion: Indicate the main conclusions or interpretations.”
1.2 Correct the following:
Line 19: terms of level of respiratory -> terms of the level of respiratory
Line 21: but still, they experienced more -> but still, experienced more
Line 23: lymphocyte to monocyte -> lymphocyte-to-monocyte
Line 26: and the aggregate index of systemic -> and aggregate index of systemic
2. Introduction:
2.1 Correct the following:
Line 40: healthcare system response -> healthcare system responses
Line 41-42: has been a challenge -> have been a challenge
Line 48: putting effort in expanding -> putting effort into expanding
Line 49: with the goal to better understand -> with the goal of better understanding
Line 52: of the COVID-19-related -> of COVID-19-related
Etc...
2.2. In the “Introduction” the authors should emphasize the negative impact of the COVID-19 Pandemic on medical activity worldwide. I suggest citing the following reference:
· https://doi.org/10.3390/jcm11092452
· https://doi.org/10.3389/fsurg.2022.883935
3. Materials and methods
3.1 I recommend the authors to divide this section into subsections that can be easily readable for the specific information (the type of study, inclusion and exclusion criteria, etc.).
3.2 Why were no patients admitted in the 3rd wave? Why was that period excluded?
3.3 Correct the following:
Line 152: especially with regards to NLR, dNLR, and SII -> especially concerning NLR, dNLR, and SII
4. Results
4.1 Please add the abbreviations below Table 1 (HGB = hemoglobin, RBC = red blood cells 76 count, RDW = ... etc.). This applies to all tables.
4.2 I suggest renaming “Combined indexes” into “Biomarkers” (for Table 1 and in the manuscript as well: Lines 151, 217)
4.3 Correct the following:
Line 145: that rose to -> that raise to
Line 187: due to their ageing -> due to their aging
5. Discussion:
5.1 I highly recommend the authors to improve the quality of the research by comparing the results with other articles in the Discussion section. Please add and compare the results with the following articles:
- https://doi.org/10.3390/diagnostics12102379
5.2. Moreover, it is important to emphasize the predictive role of the hematological ratios in cardiovascular disease, other chronic conditions, oncological etc... For example:
- https://doi.org/10.3390/life12091447
- https://doi.org/10.3390/biomedicines10061272
6. Please add a Conclusion section, after the Discussions.
Kind regards
Author Response
Response to Reviewer 1 Comments.
Point 1
- Abstract:
1.1 Please re-write the Abstract to conform to the requirements:” Background: Place the question addressed in a broad context and highlight the purpose of the study; Methods: Describe briefly the main methods or treatments applied. Include any relevant preregistration numbers, and species and strains of any animals used. Results: Summarize the article's main findings; Conclusion: Indicate the main conclusions or interpretations.”
1.2 Correct the following:
Line 19: terms of level of respiratory -> terms of the level of respiratory
Line 21: but still, they experienced more -> but still, experienced more
Line 23: lymphocyte to monocyte -> lymphocyte-to-monocyte
Line 26: and the aggregate index of systemic -> and aggregate index of systemic
Response 1:
- we have re-written the Abstract to ensure that compliance requirements are met. As per journal guidelines, we used a structured single paragraph without headings.
- We have made the corrections required.
- Introduction:
2.1 Correct the following:
Line 40: healthcare system response -> healthcare system responses
Line 41-42: has been a challenge -> have been a challenge
Line 48: putting effort in expanding -> putting effort into expanding
Line 49: with the goal to better understand -> with the goal of better understanding
Line 52: of the COVID-19-related -> of COVID-19-related
Etc...
2.2. In the “Introduction” the authors should emphasize the negative impact of the COVID-19 Pandemic on medical activity worldwide. I suggest citing the following reference:
- https://doi.org/10.3390/jcm11092452
- https://doi.org/10.3389/fsurg.2022.883935
Response 2:
2.1 all misspellings have been corrected.
2.2 thank you, we have expanded the introduction section and added a few references, including the abovementioned 10.3390/jcm11092452.
- Materials and methods
3.1 I recommend the authors to divide this section into subsections that can be easily readable for the specific information (the type of study, inclusion and exclusion criteria, etc.).
3.2 Why were no patients admitted in the 3rd wave? Why was that period excluded?
3.3 Correct the following:
Line 152: especially with regards to NLR, dNLR, and SII -> especially concerning NLR, dNLR, and SII
Response 3:
3.1 there are now 4 subsections for Materials and methods (Study design, Inclusion criteria, Data collection, Statistics), thank you for the suggestion.
3.2 The third wave was excluded because of the relatively small number of cases observed in that period in our area; we have added a clarification in the text.
3.3 corrected, thank you
- Results
4.1 Please add the abbreviations below Table 1 (HGB = hemoglobin, RBC = red blood cells 76 count, RDW = ... etc.). This applies to all tables.
4.2 I suggest renaming “Combined indexes” into “Biomarkers” (for Table 1 and in the manuscript as well: Lines 151, 217)
4.3 Correct the following:
Line 145: that rose to -> that raise to
Line 187: due to their ageing -> due to their aging
Response 4:
4.1 thank you, we have corrected all tables.
4.2 we believe it is necessary to keep the name “combined indexes” because we are not referring to just any of the analyzed biomarkers (e.g., hemoglobin, or WBC), but specifically to the CBC-derived indexes.
4.3 we have corrected ageing to aging, thank you. As for line 145, we keep the same verb tense throughout the clause for consistency
- Discussion:
5.1 I highly recommend the authors to improve the quality of the research by comparing the results with other articles in the Discussion section. Please add and compare the results with the following articles:
- https://doi.org/10.3390/diagnostics12102379
5.2. Moreover, it is important to emphasize the predictive role of the hematological ratios in cardiovascular disease, other chronic conditions, oncological etc... For example:
- https://doi.org/10.3390/life12091447
- https://doi.org/10.3390/biomedicines10061272
Response 5: We have expanded the discussion and added the article 10.3390/diagnostics12102379 to our references.
- Please add a Conclusion section, after the Discussions.
Response 6: we have added a Conclusions section.
Reviewer 2 Report
Referee report for Healthcare
Manuscript title: “Comparison of clinical characteristic, CBC–derived ratios and outcomes of hospitalized patients with COVID-19: a retrospective study of the second and fourth waves”
Manuscript ID: healthcare-1990741
This a rather descriptive small sample study. My main concerns are the study’s contribution and that fact that the statistical analysis stopped at tests between independent samples when data for much more were available to the researchers. Detailed review follows.
Lines 93-95. In general, these statistical tests for normality are too strict to the point that the most common distribution in natural phenomena—the normal distribution—becomes ultra rare. In any case, some skewness is not a problem and means with standard deviations may actually provide better information.
Table 1. It is better to provide proportions, or the data for each category, than a division. These divisions do not help the reader to compare the group characteristics. Moreover, in the Charlson index the median value is missing and a reference is necessary in the table notes or in the text. Finally, the PaO2/FiO2 ratio needs to be defined.
Lines 119-120. I invite the authors to compute the ratio using the mean values rather than the median of the Charlson index. Because the index is an integer and the values are usually small, medians are also integers and very small and may be overestimating the true ratio.
Line 151. Authors mention “the intensity of care is required”. How is this measured? Is it presented in the previous tables? Table 4 results rely on this measurement but I cannot see that variable in the text.
Discussion.
The first paragraph is a discussion of the topic and not of the research, therefore it belongs the to introduction section. The first paragraph of the discussion should summarize the research question and findings. The third paragraph is massive and needs to break in, at least, 2. I suggest a break after reference 36.
Lines 222-227. One can actually test these hypotheses with multivariate regressions rather than speculating.
Lines 230. Typo. “External VALIDITY”
Lines 230-232. The fact the sample characteristics are different between waves is presented by the authors as a strength, but in order to isolate the effect of the CDC indices on any outcome, one needs the rest of characteristics to be equally distributed between groups.
Author Response
Response to Reviewer 2 comments.
This a rather descriptive small sample study. My main concerns are the study’s contribution and that fact that the statistical analysis stopped at tests between independent samples when data for much more were available to the researchers. Detailed review follows.
Response: our study is limited by its retrospective, single center nature. The sample size represents the nearly full number of patients admitted to the respiratory unit of a district hospital serving a population of about 200,000 people during the two major epidemic waves. Please see below for detailed responses.
Lines 93-95. In general, these statistical tests for normality are too strict to the point that the most common distribution in natural phenomena—the normal distribution—becomes ultra rare. In any case, some skewness is not a problem and means with standard deviations may actually provide better information.
We thank for referee suggestion however international guidelines suggest that …”to adequately control for Type I error rate in the conditional testing procedure, sample size of at least 200 is recommended with extremely skewed populations to apply parametric tests”. This means that for sample size lesser than 200 non-parametric tests are mandatory to avoid type I error (Nguyen DP et al. Parametric Tests for Two Population Means under Normal and Non-Normal Distributions. Journal of Modern Applied Statistical Methods. May 2016, Vol. 16, No. 1, 141-159).
Table 1. It is better to provide proportions, or the data for each category, than a division. These divisions do not help the reader to compare the group characteristics. Moreover, in the Charlson index the median value is missing and a reference is necessary in the table notes or in the text. Finally, the PaO2/FiO2 ratio needs to be defined.
Response: thank you, we have made a few changes to Table 1. Data are now expressed as proportions (%) or as median IQR when necessary, so that the comparison between the two groups is easier. The missing value for Charlson Comorbidity Index in wave 2 has been corrected, and the original reference has been added in the table, thank you. We have inserted a definition for the PaO2/FiO2 ratio in the table notes.
Lines 119-120. I invite the authors to compute the ratio using the mean values rather than the median of the Charlson index. Because the index is an integer and the values are usually small, medians are also integers and very small and may be overestimating the true ratio.
Response: As above for non-normal distribution variables and parametric test application.
Line 151. Authors mention “the intensity of care is required”. How is this measured? Is it presented in the previous tables? Table 4 results rely on this measurement but I cannot see that variable in the text.
Response: the intensity of care is measured by the proportion of patients treated with oxygen only, treated with either CPAP or NIMV, and transferred to ICU, i.e., the endpoints depicted in Table 3. We have added a clarification in both text and table.
The first paragraph is a discussion of the topic and not of the research, therefore it belongs the to introduction section. The first paragraph of the discussion should summarize the research question and findings. The third paragraph is massive and needs to break in, at least, 2. I suggest a break after reference 36.
Response: Thank you, we have revised the first paragraph of the Discussion section. We have added a paragraph break after references 36 as suggested.
Lines 222-227. One can actually test these hypotheses with multivariate regressions rather than speculating.
We thank the reviewer for this comment. We fully acknowledge that to test the hypothesis that a biomarker is independently associated with any outcome, univariate and multivariate analyses against confounding factors are warranted. However, the association between blood cell-derived indexes and adverse clinical outcomes in COVID-19 has indeed already been established, and there is extensive scientific evidence in the literature including, of course, large studies with multivariable adjustments (for example, see 10.1001/jamanetworkopen.2020.22310, 10.1038/s41375-020-0911-0, 10.1016/j.mad.2022.111674). For this reason, our work was overtly focused on the description of the changing patterns of these biomarkers in a real-world setting, with two groups of patients with intrinsically heterogeneous characteristics and dissimilar outcomes.
We have modified the discussion section to highlight the descriptive aim of our article and added the above-mentioned articles to the reference list.
Lines 230. Typo. “External VALIDITY”
Response: Corrected.
Lines 230-232. The fact the sample characteristics are different between waves is presented by the authors as a strength, but in order to isolate the effect of the CDC indices on any outcome, one needs the rest of characteristics to be equally distributed between groups.
Response: see above for lines 222-227
Round 2
Reviewer 1 Report
no further comment
Author Response
We really thank the Reviewer for appreciating our work. A native speaker revised the paper before the final versions' resubmission.